# TcpC Inhibits M1 but Promotes M2 Macrophage Polarization via Regulation of the MAPK/NF-κB and Akt/STAT6 Pathways in Urinary Tract Infection

**DOI:** 10.3390/cells11172674

**Published:** 2022-08-28

**Authors:** Jiaqi Fang, Qian Ou, Boheng Wu, Sisi Li, Mian Wu, Jialing Qiu, Nuo Cen, Kaixin Hu, Yangfei Che, Yuan Ma, Jianping Pan

**Affiliations:** 1Institute of Translational Medicine, Zhejiang University City College, Hangzhou 310015, China; 2Department of Clinical Medicine, Zhejiang University City College School of Medicine, Hangzhou 310015, China; 3The First Affiliated Hospital, Zhejiang University School of Medicine, Hangzhou 310003, China

**Keywords:** UPEC, TcpC, macrophage polarization, MAPK/NF-κB, Akt/STAT

## Abstract

**Abstract:**

TcpC is a multifunctional virulence factor of Uropathogenic *Escherichia coli* (UPEC). Macrophages can differentiate into two different subsets M1 and M2 that play distinct roles in anti-infection immunity. Here, we investigate the influence of TcpC on M1/M2 polarization and the potential mechanisms. Our data showed that M1 markers CD86 and iNOS were significantly inhibited, while the M2 markers CD163, CD206 and Arg-1 were enhanced in macrophages in kidneys from the TcpC-secreting wild-type CFT073 (CFT073^wt^)-infected pyelonephritis mouse model, compared with those in macrophages in kidneys from TcpC knockout CFT073 mutant (CFT073^Δ^*^tcpc^*)-infected mice. CFT073^wt^ or recombinant TcpC (rTcpC) treatment inhibits LPS + IFN-γ-induced CD80, CD86, TNF-α and iNOS expression, but promotes IL-4-induced CD163, CD206, Arg-1 and IL-10 expression in both human and mouse macrophage cell lines THP-1 and J774A.1. Moreover, rTcpC significantly attenuated LPS + IFN-γ-induced phosphorylation of p38, ERK, p50 and p65 but enhanced IL-4-induced phosphorylation of Akt and STAT6. These data suggest that TcpC inhibits M1 but promotes M2 macrophage polarization by down-regulation of p38, ERK/NF-κB and up-regulation of the Akt/STAT6 signaling pathway, respectively. Our findings not only illuminate the regulatory effects of TcpC on macrophage M1/M2 polarization and its related signaling pathways, but also provide a novel mechanism underlying TcpC-mediated immune evasion of macrophage-mediated innate immunity.

**Simple Summary:**

We investigate the influence of TcpC, a multifunctional virulence factor of Uropathogenic *Escherichia coli* (UPEC), on M1/M2 macrophage polarization and the potential mechanisms. TcpC-secreting wild-type CFT073 (CFT073^wt^) or recombinant TcpC (rTcpC) treatment inhibits LPS + IFN-γ-induced CD80, CD86, TNF-α and iNOS expression, but promotes IL-4-induced CD163, CD206, Arg-1 and IL-10 expression in CFT073^wt^-infected pyelonephritis model mouse and both human and mouse macrophage cell lines THP-1 and J774A.1, respectively. Moreover, rTcpC significantly attenuated LPS + IFN-γ-induced phosphorylation of p38, ERK, p50 and p65 but en-hanced IL-4-induced phosphorylation of Akt and STAT6. These data suggest that TcpC inhibits M1 but promotes M2 macrophage polarization by down-regulation of p38, ERK/NF-κB and up-regulation of the Akt/STAT6 signaling pathway, respectively. Our findings not only illuminate the regulatory effects of TcpC on macrophage M1/M2 polarization and its related signaling pathways, but also provide a novel mechanism underlying TcpC-mediated immune evasion of macrophage-mediated innate immunity.

## 1. Introduction

Urinary tract infection (UTI) is urinary tract inflammation caused by various pathogens invading the urinary system, including urethritis, cystitis, pyelonephritis, etc. Some serious cases in patients can lead to renal failure or bacteremia [1,2]. It has been estimated that about 150 million people worldwide develop UTI each year, with high social costs in terms of hospitalizations and medical expenses [3,4]. Uropathogenic *Escherichia coli* (UPEC) is the most common pathogen of UTI; approximately 80% of UTIs are caused by UPEC [5,6].

Toll/interleukin-1 receptor (TIR) domain-containing protein (TcpC) encoded by the *tcpc* gene is a crucial virulence factor in most strains of UPEC [7]. The TIR domain of TcpC (TcpC-TIR) impedes the Toll-like receptor (TLR) signaling pathway by direct association with myeloid differentiation factor 88 (MyD88) and TLR4, promoting bacterial survival and increasing the severity of UTIs in human and mice [7]. Moreover, TcpC inhibits the activation of the NLRP3 inflammasome by binding to NLRP3 and caspase-1, leading to inhibition of IL-1β production in macrophages [8]. Finally, TcpC subverts the TLR-mediated nuclear factor-κB (NF-κB) signaling pathway and suppresses NETosis by serving as an MyD88- or peptidylarginine deiminase 4 (PAD4)-targeted E3 ubiquitin ligase that promotes degradation of MyD88 and PAD4 in macrophages and neutrophils, respectively [9,10]. Therefore, TcpC is a multifunctional virulence factor that inhibits innate immunity by dampening the inflammatory response.

Macrophages are inflammatory cells with a high capacity for engulfing and digesting pathogens and cell debris [11]. A number of transcriptional regulators participate in the differential activation of macrophages [12,13,14]. Macrophages are activated in response to infection, which is induced through NF-κB, mitogen-activated protein kinase (MAPK) and signal transducer and activator of transcription (STAT) signaling pathways [15,16,17,18,19]. Activated macrophages can be polarized to a proinflammatory M1 phenotype or an anti-inflammatory M2 phenotype. M1 macrophages are stimulated by interferon-γ (IFN-γ) and bacterial lipopolysaccharide (LPS) or tumor necrosis factor-alpha (TNF-α) to produce proinflammatory cytokines, such as interleukin-6 (IL-6). In contrast, M2 macrophages are induced by IL-4, IL-13 or IL-10 to express anti-inflammatory cytokines, including IL-10 [20,21]. The balance of M1 and M2 macrophages and the appropriate resolution of macrophage inflammatory response are critical for host innate immunity and adaptive immunity [22,23,24].

Although TcpC was demonstrated to play a crucial role by blocking the NF-κB pathway in the immune evasion of macrophage-mediated innate immunity [7,9], its influence on M1/M2 macrophage polarization remains unknown. Here, we show that TcpC inhibits M1 but promotes M2 macrophage polarization via the MAPK/NF-κB/STAT pathway in urinary tract infection. Our findings not only illuminate the molecular mechanisms by which TcpC regulates M1/M2 macrophage polarization, but also provide novel clues to clarify the pathogenicity of UPEC.

## 2. Materials and Methods

### 2.1. Ethics Statement

Animal experiments were performed in accordance with the National Regulations for the Administration of Experimental Animals of China (1988-002) and the National Guidelines for Experimental Animal Welfare of China (2006-398). All animal experimental protocols were approved by the Ethics Committee for Animal Experiment of Zhejiang University City College (No. 22024).

### 2.2. Cells, Bacteria Strains, Antibodies and Chemicals

The murine macrophage J774A.1 and human monocytic THP-1 cell line were purchased from the Shanghai Institute of Biochemistry and Cell Biology (Shanghai, China). The cells were cultured in RPMI 1640 medium (Gibco, Billings, MT, USA), supplemented with 10% fetal bovine serum (Gibco, Billings, MT, USA), 100 U/mL penicillin and 100 U/mL streptomycin (Sigma, Saint Louis, MO, USA), in a humidified 5% CO_2_ incubator (Thermo Fisher Scientific, Waltham, MA, USA) at 37 °C.

TcpC-expressing wild-type uropathogenic *E. coli* CFT073 strain (CFT073^wt^) was kindly provided by Professor Jian-Guo Xu (State Key Laboratory for Infectious Disease Prevention and Control, National Institute for Communicable Disease Control and Prevention, China). TcpC knockout CFT073 mutant strain (CFT073^Δ*tcpc*^) was constructed in our laboratory as described previously [25]. The strains were cultured in LB medium at 37 °C.

Primary antibodies included rabbit anti-iNOS-IgG (ab178945, Abcam, Cambridge, UK), rabbit anti-Arg-1-IgG (93668T, CST, Parsons, KS, USA), rabbit anti-ERK1/2-IgG (9102S, CST, Parsons, KS, USA), mouse anti-phospho-ERK1/2 (Tyr204/Tyr187)-IgG (5726S, CST, Parsons, KS, USA), rabbit anti-p38-IgG (YT3513, Immunoway, Plano, TX, USA), rabbit anti-phospho-p38 (T180/Y182)-IgG (YP0338, Immunoway, Plano, TX, USA), rabbit anti-JNK-IgG (9252, CST, Parsons, KS, USA), mouse anti-phospho-JNK (Thr183/Tyr185)-IgG (4668, CST, Parsons, KS, USA), mouse anti-p50-IgG (YM33100, Immunoway, Plano, TX, USA), rabbit anti-phospho-p50 (Ser337)-IgG (YP0186, Immunoway, Plano, TX, USA), rabbit anti-p65-IgG (YM3111, Immunoway, Plano, TX, USA), rabbit anti-phospho-p65 (Ser536)-IgG (YP0191, Immunoway, Plano, TX, USA), rabbit anti-Akt-IgG (4691T, CST, Parsons, KS, USA), rabbit anti-phospho-Akt (Ser473)-IgG (4060T, CST, Parsons, KS, USA), rabbit anti-STAT3-IgG (ab68153, Abcam, Cambridge, UK), rabbit anti-phospho-STAT3 (Y705)-IgG (ab267373, Abcam, Cambridge, UK), rabbit anti-STAT6-IgG (ab32520, Abcam, Cambridge, UK), rabbit anti-phospho-STAT6 (Y641)-IgG (ab263947, Abcam, Cambridge, UK) and rabbit anti-GAPDH-IgG (ab181602, Abcam, Cambridge, UK). Secondary antibodies included DyLight 800 goat anti-rabbit-IgG (RS23920, Immunoway, Plano, TX, USA) and DyLight 680 goat anti-rabbit-IgG (RS23710, Immunoway, Plano, TX, USA). Antibodies used in flow cytometry included CD80-PE monoclonal antibody (12-0801-82, eBioscience, San Diego, CA, USA), CD86-PE monoclonal antibody (12-0869-42, eBioscience, San Diego, CA, USA), CD163-PerCP monoclonal antibody (46-1639-42, eBioscience, San Diego, CA, USA) and CD206-FITC monoclonal antibody (53-2069-42, eBioscience, San Diego, CA, USA).

Chemicals and cytokines included lipopolysaccharide (LPS) (L2630, Sigma, Saint Louis, MO, USA), IFN-γ (HY-P7025, MedChemExpress, Monmouth Junction, NJ, USA) and Interleukin-4 (IL-4) (HY-P70445, MedChemExpress, Monmouth Junction, NJ, USA); LPS-free recombinant TcpC (rTcpC) was prepared in our laboratory as in previous report [9].

### 2.3. Mouse Pyelonephritis Model and In Situ M1/M2 Polarization Examination

Female C57BL/6 mice, 6–8 weeks of age, were purchased from SLC Laboratory Animal Co., Ltd. (Shanghai, China) and were housed in specific pathogen-free conditions. Mouse pyelonephritis (PN) models were prepared as per our previous report [9]. The model mice were sacrificed 3 days later, and kidneys were obtained for pathological examination. After the sections were fixed and tissue antigen repaired, the fixed tissues were incubated with FITC-F4/80, PE-CD86 and PerCP-CD163 (1:100 dilution) overnight at 4 °C in the dark, then we added ProLong Diamond Antifade Mountant (Thermo Fisher Scientific, Waltham, MA, USA) and DAPI (Sigma, Saint Louis, MO, USA) to each coverslip, then the coverslips were placed on the slides and confocal microscope images were taken.

### 2.4. Macrophage Differentiation, Polarization and Treatment

THP-1 cells were treated with 100 nM phorbol-12-myristate-13-acetate (PMA) (Sigma, Saint Louis, MO, USA) for 24 h to induce cellular differentiation into macrophages [26]. LPS (100 ng /mL)/IFN-γ (20 ng/mL) and IL-4 (20 ng/mL) were added into the corresponding wells to stimulate M1 and M2 macrophage polarization [27]. Macrophage cell lines (THP-1 and J774A.1) were separately co-cultured in transwell with CFT073^wt^ or CFT073^∆*tcpc*^ at a multiplicity of infection (MOI) of 100 for 24 h. To examine the influence of TcpC on M1 and M2 macrophage polarization, 1 × 10^6^ cells of THP-1 or J774A.1 were treated with or without 4 μg/mL rTcpC for 24 h. Supernatants were obtained for an ELISA assay and cells were collected for flow cytometry and qRT-PCR.

### 2.5. Quantitative RT-PCR

The total RNA of treated macrophages was extracted by RNAiso Plus (TaKaRa, Shiga, Japan), and cDNA was synthesized by reverse transcription using a PrimeScript RT reagent kit with gDNA Eraser (TaKaRa, Shiga, Japan). The mRNA levels of TNF-α, iNOS, IL-10 and Arg-1 in macrophages were detected by qRT-PCR, and GAPDH was set as an internal reference [9]. The Cq value was calculated by the 2^−^^∆∆^^CT^ method. The primers are listed in Appendix A.

### 2.6. ELISA and Griess Method

Expression of IL-10, TNF-α and Arg-1 in supernatants of macrophages with different treatments was measured by ELISA according to the instructions of the manufacturer (Novus, Saint Charles, MO, USA and eBioscience, San Diego, CA, USA). The production of NO was detected by the Griess method according to the instructions of the kit [28] (Promega, Madison, WI, USA).

### 2.7. Flow Cytometry

Macrophages were washed with PBS and resuspended to adjust the cell concentration to 1 × 10^6^/mL. A total of 1 × 10^5^ cells in each group were stained using isotype control antibodies or M1 marker antibodies (CD80-PE, CD86-PE) and M2 marker antibodies (CD163-PerCP, CD206-FITC) at 37 °C for 30 min. After staining, cells were washed twice with PBS, and then analyzed on a FACS Calibur (BD, Franklin Lakes, NJ, USA). Data analysis was performed on FlowJo v10.0.7 (BD, Franklin Lakes, NJ, USA) software.

### 2.8. Protein Extraction and Western Blot

Treated macrophages were lysed with RIPA lysis buffer (Beyotime, Jiangsu, China) at 4 °C for 30 min, and then were centrifuged at 12,000 rpm for 30 min. Protein concentration was determined by a BCA protein assay kit (Beyotime, Jiangsu, China). After 12% SDS-PAGE, proteins were electro-transferred onto PVDF membranes (Bio-Rad, Hercules, CA, USA). Proteins were respectively probed with rabbit anti-iNOS-IgG (1:10,000 dilution), rabbit anti-Arg-1-IgG (1:5000 dilution) and MAPK/NF-κB/STAT signal pathway-associated antibodies as primary antibodies and DyLight 800 goat anti-rabbit-IgG or DyLight 680 goat anti-rabbit-IgG (1:5000 dilution) as secondary antibodies. Images were developed using the Odyssey CLx Infrared Imaging System (LI-COR, Lincoln, NE, USA). Gray-scale values of bands were analyzed by ImageJ software.

### 2.9. Statistical Analysis

Data shown are Mean ± SD of three independent experiments. Images were analyzed by ImageJ software. Differences between the groups were assessed by two-way analysis of variance (ANOVA). *p* < 0.05 is considered to be statistically significant and *p* < 0.01 is extremely significant. All source data are provided in source data file.

## 3. Results

### 3.1. CFT073^wt^ Infection Inhibits In Vivo M1, but Enhances M2 Macrophage Polarization In Kidneys from Mouse PN Models

A mouse PN model was constructed by urethral instillation of CFT073^wt^ or CFT073^∆*tcpc*^ as described previously [9]. Abscesses in kidneys could be observed in CFT073^wt^-infected mice; no abscesses were present in kidneys from CFT073^∆*tcpc*^-infected mice (Appendix A). In addition, significantly increased infiltrates of neutrophil were observed in kidneys from the CFT073^wt^ group compared with those in kidneys from the CFT073^∆*tcpc*^ group (Appendix A). In situ M1 macrophage marker CD86 in kidneys from the PN mouse model induced by CFT073^wt^ was significantly inhibited, while the M2 macrophage marker CD163 was promoted profoundly, when compared with those in kidneys from the CFT073^∆*tcpc*^-infected group (Figure 1). These data suggest that TcpC may inhibit M1 but promote M2 macrophage polarization.

In order to further confirm the effect of CFT073^wt^ on the polarization of macrophages, kidney macrophages (K-macrophages) were sorted by flow cytometry as described in our previously published paper [9]. The cell purities were >95% when identified by the macrophage markers CD11b and F4/80 (Appendix A). CD86 (M1 marker) was suppressed but CD206 (M2 marker) was significantly promoted in K-macrophages of the CFT073^wt^ group measured by flow cytometry when compared with those in K-macrophages from the CFT073^∆*tcpc*^-infected group (Figure 2A–C). The mRNA and protein levels of the M1 and M2 markers iNOS and Arg-1 were detected in K-macrophages. The results showed that iNOS was restrained, whereas Arg-1 was promoted in the CFT073^wt^ group compared with those in the CFT073^∆*tcpc*^ group (Figure 2D–F), demonstrating that CFT073^wt^ infection inhibits in vivo M1 but promotes M2 polarization in kidneys of mice with PN.

### 3.2. CFT073^wt^ Suppresses M1 While Promoting M2 Macrophage Polarization In Vitro

Macrophage activation and polarization are crucial for inflammation and immune response. Proinflammatory M1 macrophages highly express CD80/CD86 on their surfaces, producing proinflammatory cytokines and NO. However, M2 macrophages up-regulate CD163/CD206, Arg-1 and anti-inflammatory cytokines [29,30]. Macrophage cell lines (THP-1 and J774A.1) were separately co-cultured, in the presence of LPS (100 ng/mL)/IFN-γ (20 ng/mL) or IL-4 (20 ng/mL), in transwell with CFT073^wt^ or CFT073^∆*tcpc*^ at MOI of 100 for 24 h. The expression of CD80/CD86 and CD163/CD206, as well as the production of iNOS and Arg-1, were analyzed. Our data showed that expression levels of the M1 markers CD80, CD86, NO and iNOS in CFT073^wt^-treated macrophages were significantly inhibited, while the M2 markers CD163, CD206 and Arg-1 were enhanced, when compared with those in CFT073^Δ*tcpc*^-treated macrophages (Figure 3). These results indicate that CFT073^wt^ suppresses M1 but promotes M2 macrophage polarization in vitro.

### 3.3. rTcpC Inhibits M1 but Promotes M2 Polarization in Both Human and Murine Macrophage Cell Lines

To further confirm if the aforementioned influence of CFT073^wt^ on polarization of macrophages was caused by TcpC, rTcpC was prepared and its effects on macrophage polarization were also examined. In accordance with previous reports [27,31,32], LPS + IFN-γ significantly induced the expression of M1 markers CD80 and CD86, and IL-4 treatment promoted the expression of M2 markers CD163 and CD206 in both THP-1 and J774A.1 macrophage cell lines. However, 4 μg/mL rTcpC treatment profoundly attenuated LPS + IFN-γ=induced expression of CD80 and CD86 (Figure 4A–D), while promoting CD163 and CD206 expression (Figure 4E–H). Moreover, rTcpC significantly inhibited LPS + IFN-γ-induced expressions of NO, TNF-α (Figure 5A,B) and iNOS (Figure 5E,F), while enhancing IL-4-induced production of IL-10 (Figure 5D) and Arg-1 (Figure 5C,E,F) in macrophages. When the mRNA levels were measured, results mirroring the same trend were also obtained (Appendix A). These data confirmed that rTcpC attenuated M1 but enhanced M2 macrophage polarization.

### 3.4. rTcpC Regulates M1/M2 Macrophage Polarization via Regulating MAPK/NF-κB and Akt/STAT Signaling Pathway

The MAPK/NF-κB/STAT pathway plays a crucial role in macrophages, which can be affected by a variety of factors to change their phenotype and, thus, affect their function [15,16,19,33]. In line with previous reports [34,35,36], LPS + IFN-γ treatment significantly resulted in phosphorylation of ERK, p38, JNK, p50 and p65 (Figure 6A,C), and IL-4 treatment led to phosphorylation of Akt, STAT3 and STAT6 in THP-1 (Figure 7A), demonstrating the activation of the MAPK-NF-κB pathway or the Akt/STAT pathway in M1 or M2 macrophage polarization, respectively. However, in the presence of rTcpC (LPS + IFN-γ + rTcpC group), the LPS + IFN-γ-induced phosphorylation of ERK, p38, p50 and p65, but not JNK, was profoundly attenuated (Figure 6A,C). Meanwhile, the IL-4-induced phosphorylation of Akt and STAT6 was enhanced significantly after treatment with rTcpC (Figure 7A). Furthermore, when the mouse macrophage cell line J774A.1 was used, results with the same trend were also obtained (Figure 6B,D and Figure 7B). These data demonstrate that rTcpC inhibits M1 polarization via blocking the ERK/p38/NF-κB pathway and promotes M2 polarization via activating the Akt/STAT6 pathway.

## 4. Discussion

Pathogens are aggressors, and host immune cells such as macrophages and neutrophils are responsible for defense. UTIs are common infections, induced predominantly by UPEC, which cause severe pyelonephritis with significantly increased infiltrates of neutrophils [37,38]. We and others reported that UPEC CFT073 strains cause kidney abscesses TcpC-dependently in mice that contain an accumulation of neutrophils and bacteria, but inhibit the activation of NF-κB to attenuate inflammatory reaction in macrophages [7,9,10,39]. During the process of pathogen infection, the tissue cells could secrete chemokines to recruit macrophage infiltration to eliminate the pathogen [40]. Then, the infiltrated macrophages could also secrete chemokines to recruit other innate immune cells including neutrophils, which secrete inflammatory factors to kill pathogens together. As an important virulence factor, TcpC plays a crucial role in the pathogenecity of UPEC. TcpC-expressing UPEC cause death and tissue damage in normal hosts by creating a dysfunctional innate immune response [41]. In addition, TcpC promotes kidney cells to produce CXCL2 chemokine, which attracts neutrophil infiltration in kidneys [25]. Therefore, the inhibition of proinflammatory factors and enhanced neutrophil infiltration in CFT073^wt^-infected mice kidneys are consistent with the findings which have been reported.

Macrophage M1/M2 polarization have been proposed to play important roles in many diseases, including inflammatory diseases, various cancers and tissue repair [42,43,44,45,46]. Specific factors can cause M2 macrophage differentiation into four subtypes: M2a, M2b, M2c and M2d—IL-4 or IL-13 for M2a, TLR agonists or IL-1 receptor ligands for M2b, IL-10 or glucocorticoids for M2c, and IL-6 or TLR agonists for M2d [47,48]. M2a macrophages are identified by cell-surface markers CD163/CD206 [49] and play a significant role in anti-inflammatory mechanisms [29]. In our research, rTcpC treatment profoundly promoted expression of CD163 and CD206 in THP-1 and J774A.1 and stimulated differentiation into M2 macrophages with IL-4, demonstrating that TcpC promotes M2a macrophage polarization, hereby favoring UPEC to escape from the macrophage-mediated inflammatory anti-infection response. The polarization of macrophages is regulated during the infection of some pathogenic microorganisms including *Cryptococcus neoformans* [50,51], *Mycobacterium tuberculosis* [52], *Staphylococcus aureus* [32], influenza virus [53] and SARS-CoV-2 virus [54]. In addition, some factors have been reported to regulate macrophage polarization, such as RORα, ZIP9, TSC and p190RhoGEF, and inhibit M1 macrophage polarization [34,55,56,57], while YAP restrains M2 macrophage polarization [49].

TcpC is a multifunctional virulence factor of UPEC that subverts the host innate immunity. TcpC inhibits phagocytic and bactericidal activity of macrophages [7,9]; it also attenuates the anti-infection activity of neutrophils by suppressing NETosis [10]. However, the regulatory effects of TcpC on macrophage M1/M2 polarization have been remained elusive. In the present study, we showed that TcpC-secreting UPEC CFT073^wt^ infection inhibits in situ M1 but promotes M2 polarization in a murine PN model (Figure 1 and Figure 2). We also showed that CFT073^wt^ and rTcpC treatment inhibits in vitro M1 and promotes M2 polarization in both human and murine macrophage cell lines (Figure 3, Figure 4 and Figure 5). These data demonstrate that TcpC-secreting CFT073^wt^ subverts macrophage polarization, which might contribute to the pathogenicity of UPEC.

The MAPK signaling pathway regulates many important cellular physiological and pathological processes, such as cell growth, differentiation and inflammatory response [58]. NF-κB, one of the major inflammatory regulators, is another key transcription factor that promotes transcription of genes encoding proinflammatory cytokines [59]. The STAT pathway is an evolutionarily conserved signal transduction paradigm, providing mechanisms for rapid receptor-to-nucleus communication and transcription control [60].

It was reported that complete polarization of macrophages to the M1 type requires activation of the MAPK and NF-κB signaling pathways [19,27,31,61], and M2 macrophage polarization requires activation of the STAT signaling pathway [62,63]. To explore the signal transduction pathways underlying the regulatory effects of TcpC on M1/M2 polarization, the influence of TcpC on MAPK/NF-κB and Akt/STAT pathways was examined. Our data clearly showed that TcpC significantly attenuated the LPS + IFN-γ-induced phosphorylation of ERK, p38, p50 and p65 (Figure 6) and enhanced the IL-4-induced phosphorylation of Akt and STAT6 in both human and murine macrophage cell lines (Figure 7), demonstrating that TcpC inhibits M1 but promotes M2 polarization by regulating MAPK/NF-κB and Akt/STAT6 pathways. In our experiments, however, phosphorylation of JNK, one of the three members in MAPK family, remained unchanged after TcpC treatment (Figure 6). The exact mechanism by which TcpC regulates the activation of different members of the MAPK family deserves further examination. Some studies have found that Lys63-linked polyubiquitination involves feedback mechanisms in control of JNK activity [64], and this modification does not cause its proteasomal degradation [65]. We have previously demonstrated that TcpC can also serve as an E3 ubiquitin ligase that promotes ubiquitination and degradation of MyD88 in macrophages, hereby suppressing the TLR-mediated innate immunity [9]. Therefore, whether TcpC promotes Lys63-linked ubiquitination of JNK and the role of JNK in TcpC regulation of M1/M2 polarization deserve further examination.

In summary, we demonstrated for the first time that TcpC suppresses M1 but promotes M2 macrophage polarization by down-regulation of ERK and p38/NF-κB and up-regulation of the Akt/STAT6 signaling pathway, respectively, hereby favoring UPEC to escape from the macrophage-mediated inflammatory anti-infection response (Figure 8). On the one hand, our findings not only illuminate the regulatory effects of TcpC on macrophage M1/M2 polarization and its related signaling pathways, but also provide a novel mechanism underlying TcpC-mediated immune evasion of macrophage-mediated innate immunity. On the other hand, our findings provide further evidence suggesting that TcpC is a potential target for clinical therapy of UTIs caused by TcpC^+^ UPEC.

On stimulation by TLR4 ligands, receptor tyrosine kinase or MyD88 recruit downstream signal molecules, leading to the activation of the MAPK/NF-κB pathway, while activation of the Akt/STAT pathway was stimulated by IL-4. Under the circumstance of TcpC-secreting UPEC infection, TcpC down-regulates ERK and p38/NF-κB and up-regulates the Akt/STAT6 signaling pathway, respectively, to inhibit M1 but promote M2 macrophage polarization, hereby favoring UPEC to escape from the macrophage-mediated inflammatory anti-infection response.

## Figures and Tables

**Figure 1 cells-11-02674-f001:**
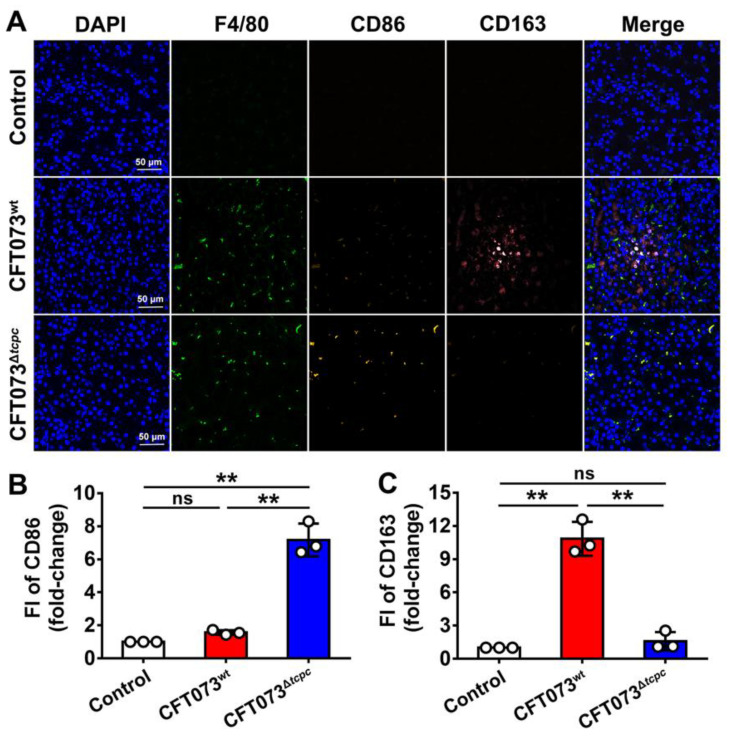
CFT073^wt^ infection inhibits CD86 but promotes CD163 in situ expression on macrophages in kidneys from PN mouse model. (**A**) The in situ expression of CD86 and CD163 on the surface of macrophages in kidneys from mouse PN model was observed by fluorescence microscope. Scale bar = 50 μm. One representative image of kidneys from 3 mice in each group was shown. (**B**) The fluorescence intensity (FI) analysis of CD86 in experiments shown in A. (**C**) FI of CD163 levels in experiments shown in A. Mean ± SD of data from three kidneys in each group are shown, *n* = 3. FI in the control group was set as 1.0. **: *p* < 0.01. ns: not significant.

**Figure 2 cells-11-02674-f002:**
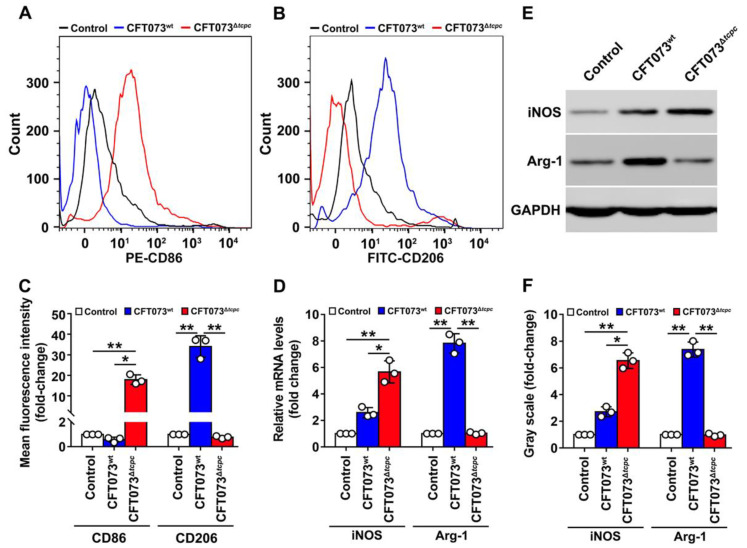
CFT073^wt^ infection significantly inhibits CD86 and iNOS but promotes CD206 and Arg-1 expression in K-macrophages from PN mouse models. (**A**,**B**) Flow cytometry analyses to examine the expression of CD86 and CD206 on K-macrophages. (**C**) FI of CD86 and CD206 were analyzed by FlowJo v10.0.7 software. FI in the control group was set as 1.0. (**D**) mRNA levels of iNOS and Arg-1 in K-macrophages were detected by qRT-PCR. (**E**) Protein levels of iNOS and Arg-1 in K-macrophages were detected by Western blot. (**F**) Gray-scale analyses of bands reflecting iNOS and Arg-1 protein levels in experiments as described in E. Gray scale in the control group was set as 1.0, Mean ± SD of three independent experiments are shown. *: *p* < 0.05; **: *p* < 0.01.

**Figure 3 cells-11-02674-f003:**
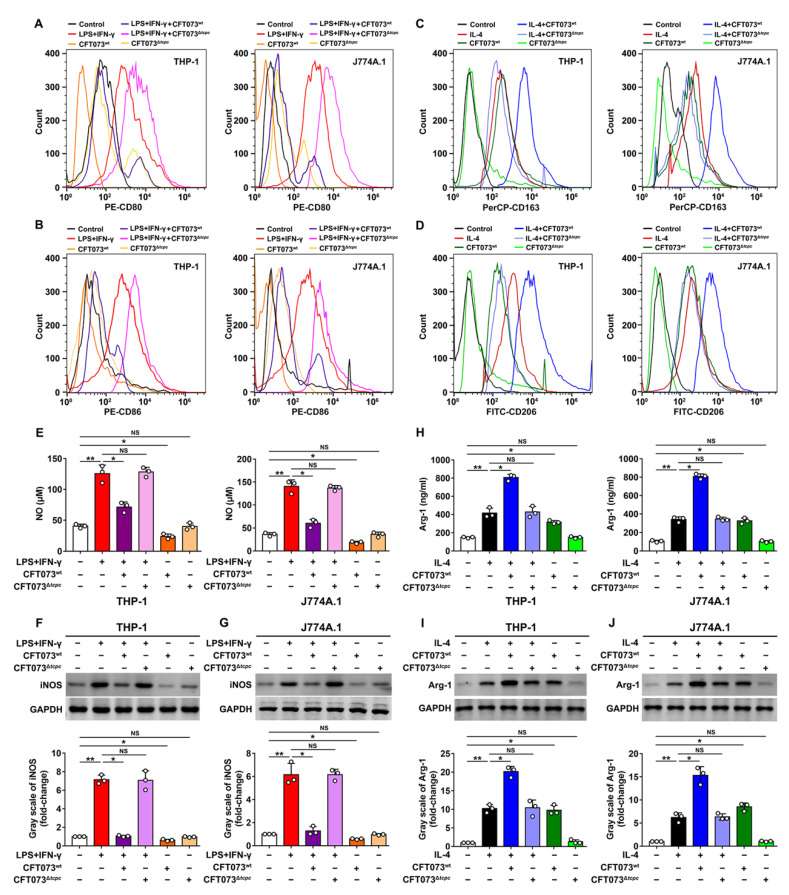
CFT073^wt^ suppresses M1 but promotes M2 macrophage polarization in vitro. (**A**,**B**) The expression of M1 surface markers CD80 and CD86 in THP-1 and J774A.1 was detected by flow cytometry. (**C**,**D**) The expression of M2 surface markers CD163 and CD206 in THP-1 and J774A.1 was analyzed by flow cytometry. (**E**) The production of NO in THP-1 and J774A.1 was measured by Griess method. (**F**,**G**) The iNOS protein levels and bands’ gray-scale analyses in THP-1 and J774A.1, respectively. (**H**) The protein levels of Arg-1 in THP-1 and J774A.1 were detected by ELISA. (**I**,**J**) The protein levels of Arg-1 were detected by Western blot and bands’ gray-scale analyses in THP-1 and J774A.1, respectively. Gray scale in the control group was set as 1.0. Mean ± SD of three independent experiments are shown. *: *p* < 0.05; **: *p* < 0.01. NS: not significant.

**Figure 4 cells-11-02674-f004:**
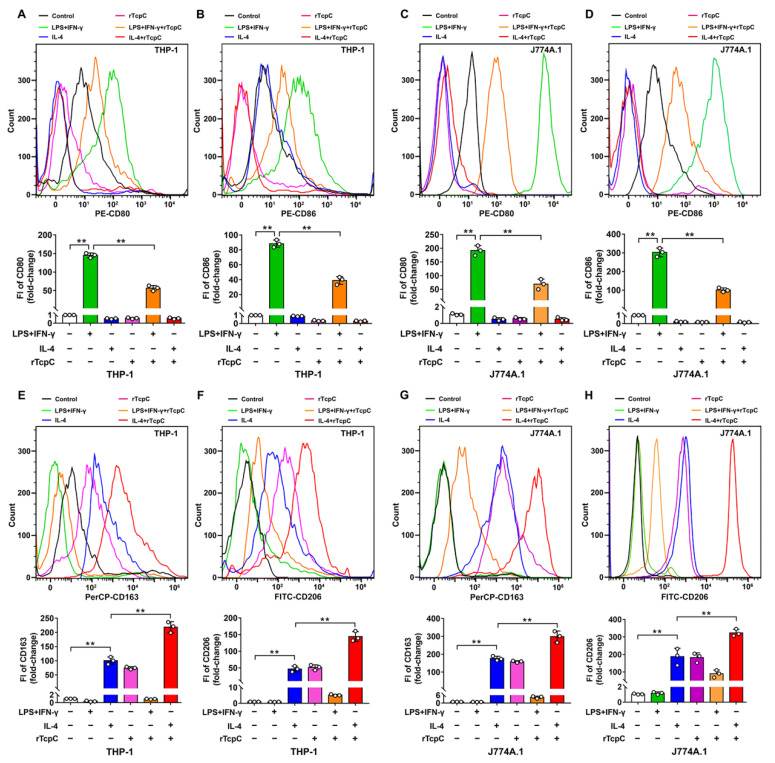
rTcpC inhibits M1 markers CD80 and CD86 but promotes the expression of M2 markers CD163 and CD206 in macrophages. (**A**–**D**) The effects of rTcpC on M1 surface markers CD80 and CD86 detected by flow cytometry and FI were analyzed by FlowJo v10.0.7 software. (**E**–**H**) The surface markers CD163 and CD206 of M2 polarized macrophages were detected by flow cytometry. Mean ± SD of three independent experiments are shown. **: *p* < 0.01.

**Figure 5 cells-11-02674-f005:**
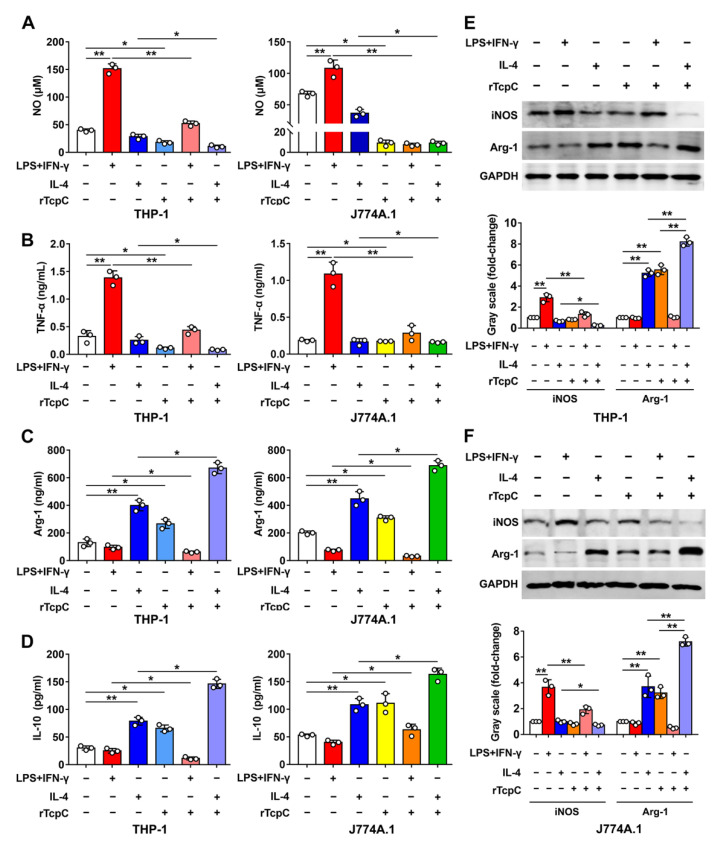
rTcpC suppresses the expression of proinflammatory factors and iNOS, but enhances the production of IL-10 and Arg-1 in macrophages. (**A**) The influences of rTcpC on NO in THP-1 and J774A.1 were detected by Griess method. (**B**–**D**) The expression levels of TNF-α, Arg-1 and IL-10 in THP-1 and J774A.1 were detected by ELISA. (**E**,**F**) The expression levels of iNOS and Arg-1 and bands’ gray-scale analyses in THP-1 and J774A.1, respectively. Gray scale in the control group was set as 1.0, Mean ± SD of three independent experiments are shown. *: *p* < 0.05; **: *p* < 0.01.

**Figure 6 cells-11-02674-f006:**
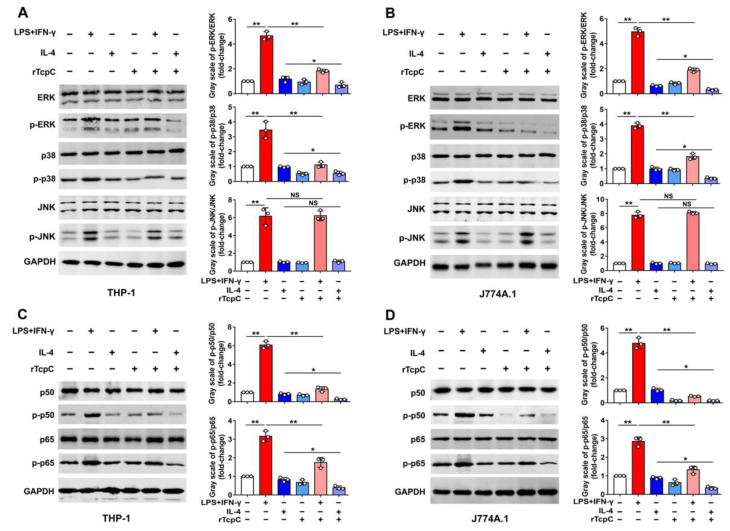
rTcpC inhibits the activation of MAPK/NF-κB pathway in macrophages. (**A**,**B**) Western blot to detect the effects of rTcpC on phosphorylation of ERK, p38, and JNK and bands’ gray-scale analyses in THP-1 and J774A.1, respectively. (**C**,**D**) Western blot analyses of phosphorylation of p50 and p65 and bands’ gray-scale analyses in THP-1 and J774A.1, respectively. Mean ± SD of three independent experiments are shown. *: *p* < 0.05; **: *p* < 0.01. NS: not significant.

**Figure 7 cells-11-02674-f007:**
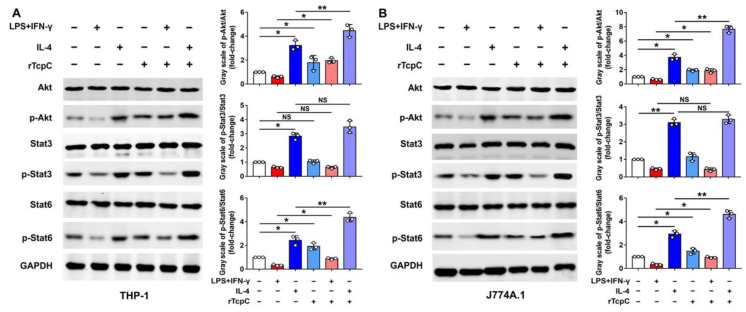
rTcpC promotes the activation of Akt/STAT pathway in macrophages. (**A**) Western blot analyses of the effects of rTcpC on phosphorylation of Akt, STAT3 and STAT6 and bands’ gray-scale analyses in THP-1. (**B**) Western blot to examine the influences of rTcpC on phosphorylation of Akt, STAT3 and STAT6 and bands’ gray-scale analyses in J774A.1. Mean ± SD of three independent experiments are shown. *: *p* < 0.05; **: *p* < 0.01. NS: not significant.

**Figure 8 cells-11-02674-f008:**
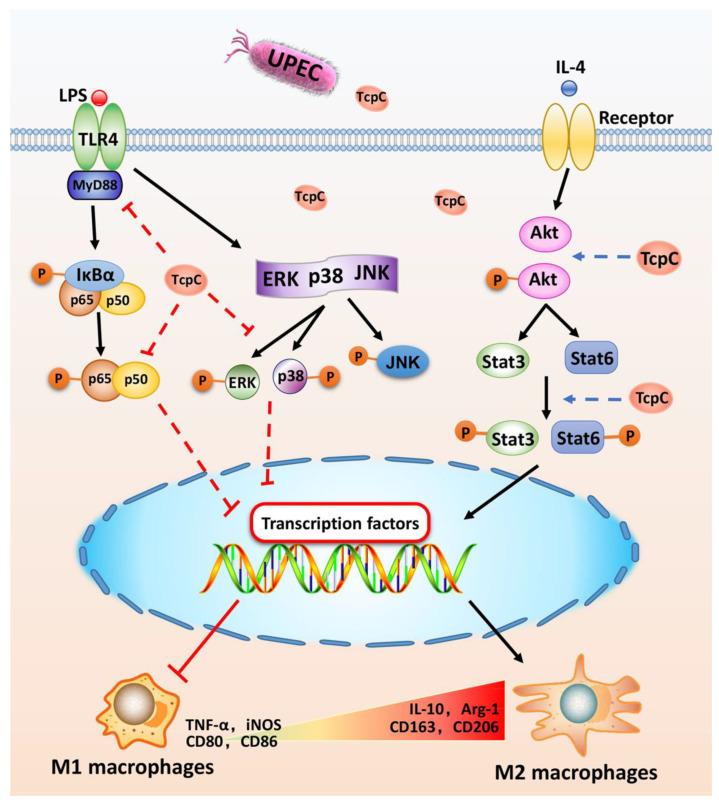
Schematic diagram of the molecular mechanisms by which TcpC inhibits M1 but promotes M2 polarization.

## Data Availability

Data supporting the findings of this manuscript are available from the corresponding author upon request. Full-scan images of the Gels, Blots and numbers are provided with this paper.

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
