# Peer review of "TcpC Inhibits M1 but Promotes M2 Macrophage Polarization via Regulation of the MAPK/NF-κB and Akt/STAT6 Pathways in Urinary Tract Infection"

_cells, 2022, doi:10.3390/cells11172674_

Round 1

Reviewer 1 Report

1. UTIs are common infections induced predominantly by UPEC, which cause severe pyelonephritis with significantly increased infiltrates of neutrophils. The author reported that TcpC inhibits M1 but promotes M2 macrophage polarization via the MAPK/NF-κB/STAT pathway in urinary tract infection. They not only illuminate the molecular mechanisms by which TcpC regulates M1/M2 macrophage polarization, but also provide novel clues to clarify the pathogenicity of UPEC. I have some suggestions for minor revision.

1. M2 macrophages are divided into M2a, M2b, M2c, and M2d subcategories. These macrophages differ in their cell surface markers, secreted cytokines and biological functions. What is the role between the different subtypes in your research?

2. The author mentioned "TcpC down-regulates ERK, p38/NF-κB and up-regulates Akt/STAT6 signaling pathway respectively to inhibit M1 but promote M2 macrophage polarization, hereby favoring UPEC to escape from the macrophage-mediated inflammatory anti-infection response."  Can the importance of TcpC result in more severe infection and even death in mice?

3. In (Figure 6A, C), the presence of rTcpC (LPS+IFN+rTcpC group) induced phosphorylation of ERK, p38, p50, p65, but not JNK was profoundly attenuated. How to explain the effect  of JNK and what can be the potential impact of M1?

4. From bacterial evading from urethra to kidney, UPEC may need several virulence factor to induce kidney infection, abscess or bacteremia. What clinical impact do you think for TcpC?

Reviewer 2 Report

The authors investigated the influence of TcpC on M1/M2 polarization and the potential mechanisms in this manuscript. By performing the Mouse PN model, the author found that TcpC inhibited M1 but promoted M2 macrophage polarization in vivo. In vitro stimulation assay proved that CFT073wt suppressed CD80, CD86, inos, and TNFa but promoted CD163, CD206, and Arg1 in THP-1 and J774A.1 cells. Although it was known that TcpC regulated NF-kB pathway, which is known to be involved in macrophage polarization, the authors found that TcpC affected MAPK, and AKT pathways. 

The manuscript was well written and easy to follow and the data and evidence were clear and sufficient to support the conclusion.
